# Study on the effect of water content on physical properties of bentonite

Chao Zheng, Yanzhao Yuan *

Lecturer College of Civil and Traffic Engineering, Henan University of Urban Construction, Ping Dingshan, China

* yyz@huuc.edu.cn

## Abstract

Moisture content profoundly influences the engineering properties of expansive soil, a critical consideration in various geotechnical applications. This study delves into the intricate relationship between water content and the physical properties of bentonite, a key constituent of expansive soil. Through a comprehensive analysis encompassing fundamental physical properties, rheological characteristics, permeability behavior, and microscopic features, we elucidate the complex interplay between water content and bentonite behavior. Our investigation reveals distinct responses to varying moisture levels: at low water content (w = 50%), unsaturated samples undergo incremental density increases attributed to moisture accumulation among particles. Concomitantly, heightened pressure fosters enhanced cohesion between particles, bolstering mechanical properties and augmenting reverse osmosis capacity. Conversely, at higher water content levels (w > w saturated), the escalation of free water within soil particles triggers pronounced particle softening, overshadowing expansion effects. Consequently, cohesion diminishes, and particles exhibit micro-scale flocculation. These findings offer valuable insights into bentonite behavior under differing moisture regimes, thereby providing a robust theoretical foundation for projects requiring bentonite seepage control.

**Data Availability Statement:** All relevant data are within the paper and its Supporting information files.

**Funding:** The author(s) received no specific funding for this work.

## 1. Introduction

Bentonite, an anti-seepage protection material [1], is widely used in various projects. For example, compacted bentonite can be used as the ideal buffer backfill material for the nuclear waste repository. At the same time, powder sprayed bentonite can be used as the primary anti-seepage layer material for flexible curtains [2], anti-seepage and leakage removal material for reservoir bottom anti-seepage cushion material for landfills and anti-seepage blanket for water conservancy projects. Therefore, studying the fundamental properties of bentonite can effectively improve the utilisation efficiency for different projects [3, 4].

Presently, research on bentonite properties mainly focuses on the anti-seepage performance and the strength of the bentonite mixture. Different scholars have studied the change rule of anti-seepage interpretation and the resilience of bentonite mixture under various influencing factors (such as initial water content, initial dry density, mineral composition, ph value, etc.)

**Competing interests:** The authors have declared that no competing interests exist.

[3, 4]. However, some scholars have found that with the deepening of water content, the physical properties of bentonite change greatly [5, 6], and the evolution of water content of bentonite is essential for the project using bentonite as the primary anti-seepage material [7].

Given the findings from previous research, whether bentonite is used as a seepage prevention material in nuclear engineering or as a soil treatment material in civil engineering for foundation improvement, the changing patterns of bentonite's physical properties upon contact with water have consistently been a primary focus. The physical characteristics of bentonite are intricately linked to its water content. On one hand, the cohesion and internal friction angle of bentonite exhibit complex changes as water content increases. On the other hand, the expansion and deformation of particles with rising water content further impact the water-holding capacity of bentonite, resulting in notable variations in its water absorption performance. Therefore, studying how water affects the physical properties of bentonite is particularly crucial. Based on the research results on the physical properties of bentonite in Nanyang, China, under different water contents, this paper will focus on the fundamental physical properties of bentonite and use a rheometer to measure the fluid properties of bentonite and use a stereomicroscope to comprehensively analyse the influence of water contents on physical properties of bentonite under different water contents, which will provide an experimental basis for construction engineering and constitutive research of bentonite.

## 2. Experimental study

The experimental investigation involved tests on bentonite mud to evaluate its free swelling behavior, penetration characteristics, rheological properties, and microscopic features under different hydration conditions. Bentonite samples with varying water content were prepared, and a series of tests were conducted to assess their physical and mechanical properties.

### 2.1 Material

Bentonite obtained from Shandong, China, characterized by high montmorillonite content and fine particle size, was used in the experiments. Chemical composition analysis revealed its suitability for the intended study.shown in Table 1.

### 2.2 Free swelling test

Samples of bentonite powder with different water content were subjected to free swelling tests to measure their expansion behavior. The expansion of the samples over time provided insights into the hydration process and the influence of water content on swelling characteristics.

### 2.3 Rheological tests

The rheological properties of bentonite mud were evaluated using a rheometer to measure parameters such as yield stress, plastic viscosity, and thixotropy [6, 7]. The rheological behavior of bentonite under varying hydration conditions was analyzed to understand its fluid properties.analysis's software can control torque (230 mNm) and measure rotational speed (314 rad/s). Computation of shear stress and shear rate. The ratio determines the apparent viscosity of the suspension.

**Table 1. Chemical component.**

| Element | $SiO_2$ | $TiO_2$ | $Al_2O_3$ | $Fe_2O_3$ | FeO | MnO | CaO | MgO | $Na_2O$ | $K_2O$ | loss |
|---|---|---|---|---|---|---|---|---|---|---|---|
| Percentage (%) | 70.12 | 0.25 | 12.19 | 0.12 | 0.38 | 0.07 | 0.52 | 0.54 | 2.16 | 0.81 | 5.12 |

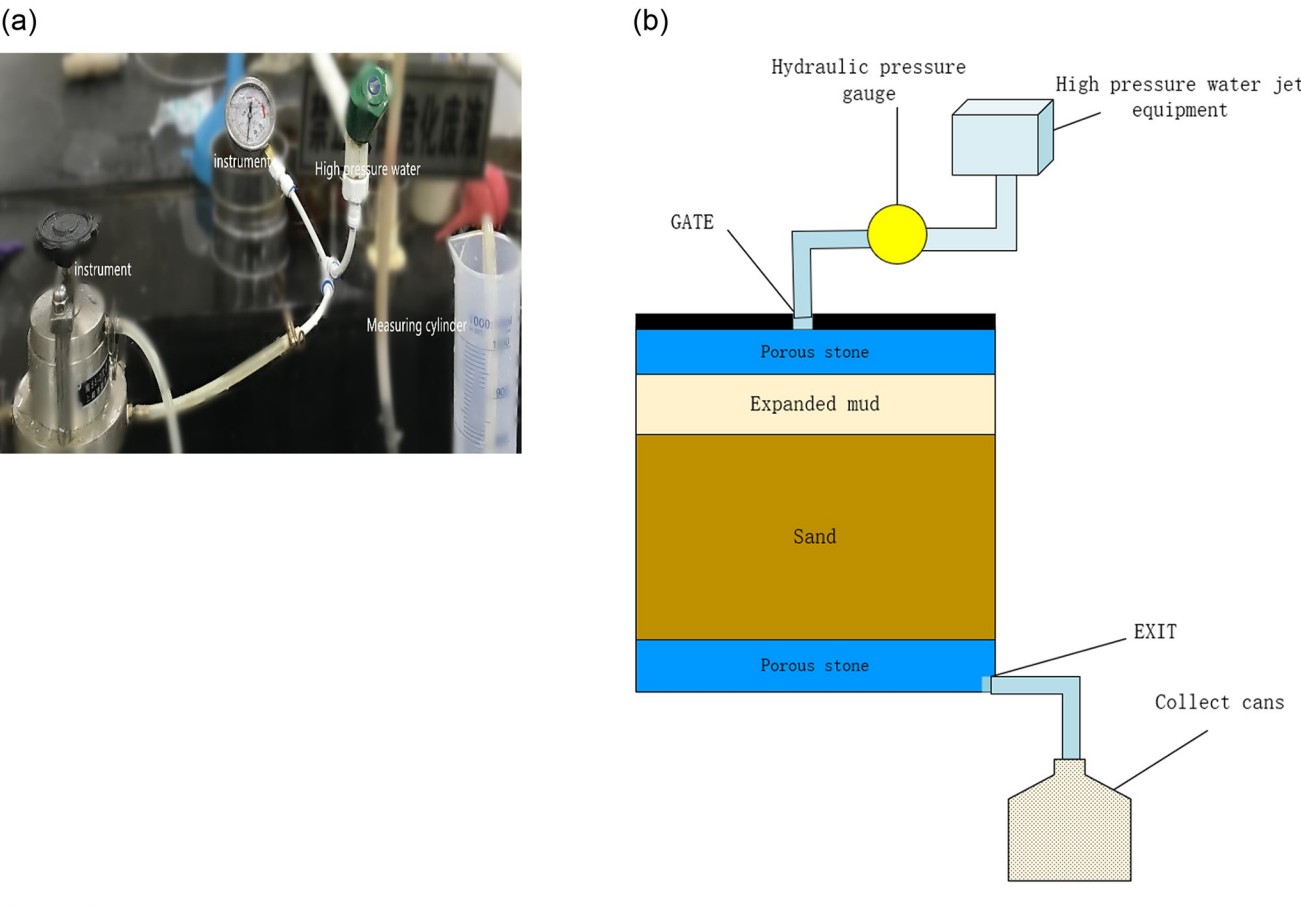

**Fig 1. Drain test.**

## 2.4 Permeability test

Permeability tests were conducted to assess the ability of bentonite to resist seepage under different pressure conditions. The experiments involved injecting mud under varying pressures to simulate real-world scenarios and analyze the hydration anti-seepage effect of the bentonite slurry. as shown in Fig 1.

The consolidation instrument was divided into two layers, the bottom sand layer was 30mm, and the upper slurry layer was 10mm. The water of different pressure is injected into the upper slurry layer, and the slurry is injected into the sand layer by high-pressure water.

According to the seepage situation of the seepage port, when there is no seepage, the pressure increases by 0.2 Mpa every 5 minutes. When the percolating water became linear, the pressure was maintained until the volume of percolating water reached 1000 ml.

## 2.5 Microstructure tests

Microscopic analysis of bentonite samples under different hydration conditions provided insights into the structural changes occurring at the particle level. Observations at different magnifications revealed the hydration process and the formation of cohesive structures within the bentonite matrix. The flow mechanism of expansive soil slurry in the sand layer has been elucidated by Mirza [8].

## 3. Results

The experimental results yielded valuable insights into the behavior of bentonite under varying moisture levels. The performance tests, including free expansion tests, rheological tests, permeability tests, and microstructure analysis, provided comprehensive data on the physical and mechanical properties of bentonite under different hydration conditions.

### 3.1. Performance test

Analysis of the sample grouping and density-moisture ratio revealed significant variations in the properties of bentonite mud under different water content conditions. The expansion behavior of bentonite mud was characterized by distinct stages, highlighting the influence of hydration time on swelling characteristics.

**3.1.1 Sample grouping.** The conclusions of the relevant literature configure the slurry. The density is 1.3–1.6g / cm$^3$, and the mud-water ratio of the mud test is between 0.3–1.45. Table 1 shows the proportion of the expansive soil slurry.

The sizing agent is prepared according to the conclusion of the relevant literature. The density is 1.3–1.6 g/cm$^3$, and the water content in the mud test is 0.3–1.45. As the Table 2 shows the ratio of the expansive soil mud.

**3.1.2 Analysis of the ratio of mass to density.** It was found that the proportion of groups 1–3 was solid, and the groups 4–6 were pulping. Groups 7–10 were cloudy liquids. The water-soil mixing curve can be analysed. The maximum density increase reaches 2.8 g/cm$^3$, the lower density difference is 1.3 g/cm$^3$, and the most stable density difference between the fifth and sixth groups is only 0.2 g/cm$^3$. According to the relevant density values, the fifth and sixth groups are the most stable mud meeting the requirements. The results are shown in Fig 2.

### 3.2 Free expansion tests

Free expansion test data of bentonite mud are shown in Fig 3, and the expansion curve can be divided into three stages. The growth zone, the development zone and the slow zone, in which the growth zone is the initial stage, with the rapid expansion. In contrast, the growth zone has a uniform rate and expansion rate, while the slow zone gradually eliminates expansion. The longer the development cycle of bentonite mud is, the more thorough its hydration is. Dates of hydration time showed the axial shrinkage strains are not directly measured curves, the five group's development zone is longer than other groups, the bentonite slurry, and increasing thermal conductivity with increasing sand percentages becomes stable.

### 3.3 Rheological tests

Rheological analysis showed that the rheological properties of bentonite mud varied significantly with changes in water content. As the water content increased, the mud transitioned from a solid to a liquid state, resulting in a sharp decrease in shear stress and viscosity. The relative density of the mud also influenced its rheological behavior, with higher densities leading to increased apparent viscosity. Fig 4 shows the rheological results of ten suspensions.

**Table 2. Water cement ratio of slurry.**

| Group | 1 | 2 | 3 | 4 | 5 | 6 | 7 | 8 | 9 | 10 |
|---|---|---|---|---|---|---|---|---|---|---|
| Soil-water ratio | 1:9 | 2:8 | 3:7 | 4:6 | 5:5 | 6:4 | 7:3 | 8:2 | 9:1 | 10:0 |

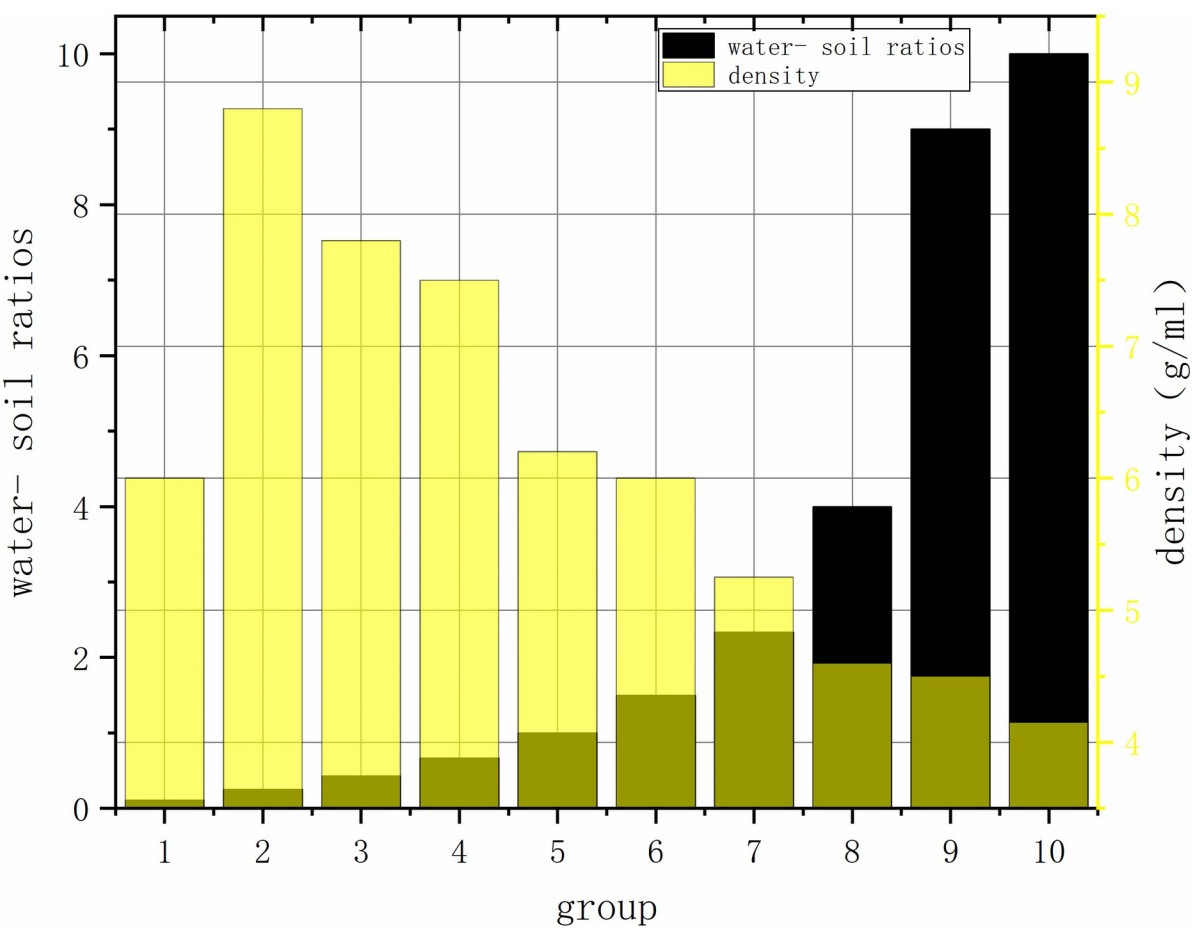

**Fig 2. The proportion of the sample.**

1. Compare rheological results of the 4 group and 5 groups; as the expansive soil slurry gradually takes the form of a liquid, the shear stress of its viscosity decreases sharply, which falls from 1.4 KPa to 2.0 KPa.

2. Based on the analysis of rheological experiments, the related tests of bentonite in longitudinal comparison can find that with the increase of water content, the slurry gradually dissipates, and the pressure-bearing capacity of the slurry decreases rapidly, and the fluidity gradually increases.

3. The relative density greatly influences the test results. At the same strain rate, with the increase of the relative density, the apparent dynamic viscosity-super porous pressure ratio curve increases gradually; Under the same ratio of super pore pressure, the pronounced dynamic viscosity-strain rate curve gradually increases.

4. Apparent dynamic viscosity decreases with the increase of strain rate, showing shear thinning characteristics. This feature exists in different pore pressure ratios, but after liquefaction, the apparent dynamic viscous train rate curve becomes steeper with the decrease of the pore pressure ratio. This change also shows that bentonite mud is traditionally hydrated, which will affect the cohesiveness of the sandy soil mixture [9–11].

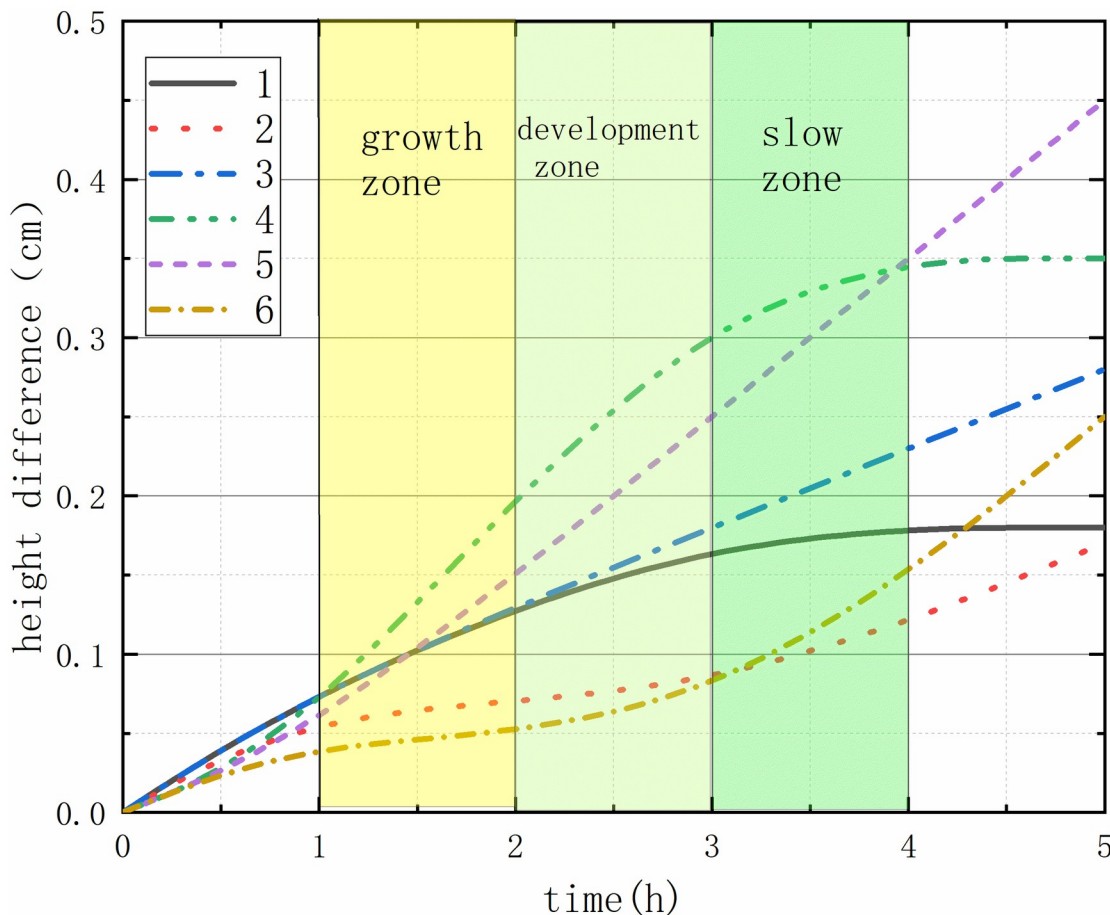

**Fig 3. The expansion height.**

### 3.4 The permeability test

Permeability tests under different pressure conditions demonstrated the efficacy of bentonite in resisting seepage. The experiments revealed a gradual decrease in permeability with increasing pressure, indicating the ability of bentonite to form an effective anti-seepage barrier under hydraulic loading. From Fig 5, it was found that the flow time reaches 1000 ml with the increase of pressure. It becomes longer gradually; even when the final high pressure is 0.26 MPa, it will take nearly 30 minutes to fill, and the whitening and grounding time is relatively late. Under the experimental conditions of the maximum pressure of 0.26 MPa, the difference between the two is 10 min, and it is between 0.18–0.22 MPa. Under the condition of seepage expansion, the seepage velocity is relatively small. From the experimental results, it can be seen that with the increase of pressure, serious cracking marks appear. The density of raffinate. The tendency of the retentate gradually increased from the soil-water ratio, and the density of slurry loss increased sharply. In the five groups, the thickness of retentate was higher and distributed linearly, while the density of retentate decreased.

 During the process of infiltration, bentonite mud is injected into the sand. With the increased water pressure, this paper records the loss of bentonite in the samples under different pressures for every 1000 ml of water seeped. Analysis of the density of the expansive soil

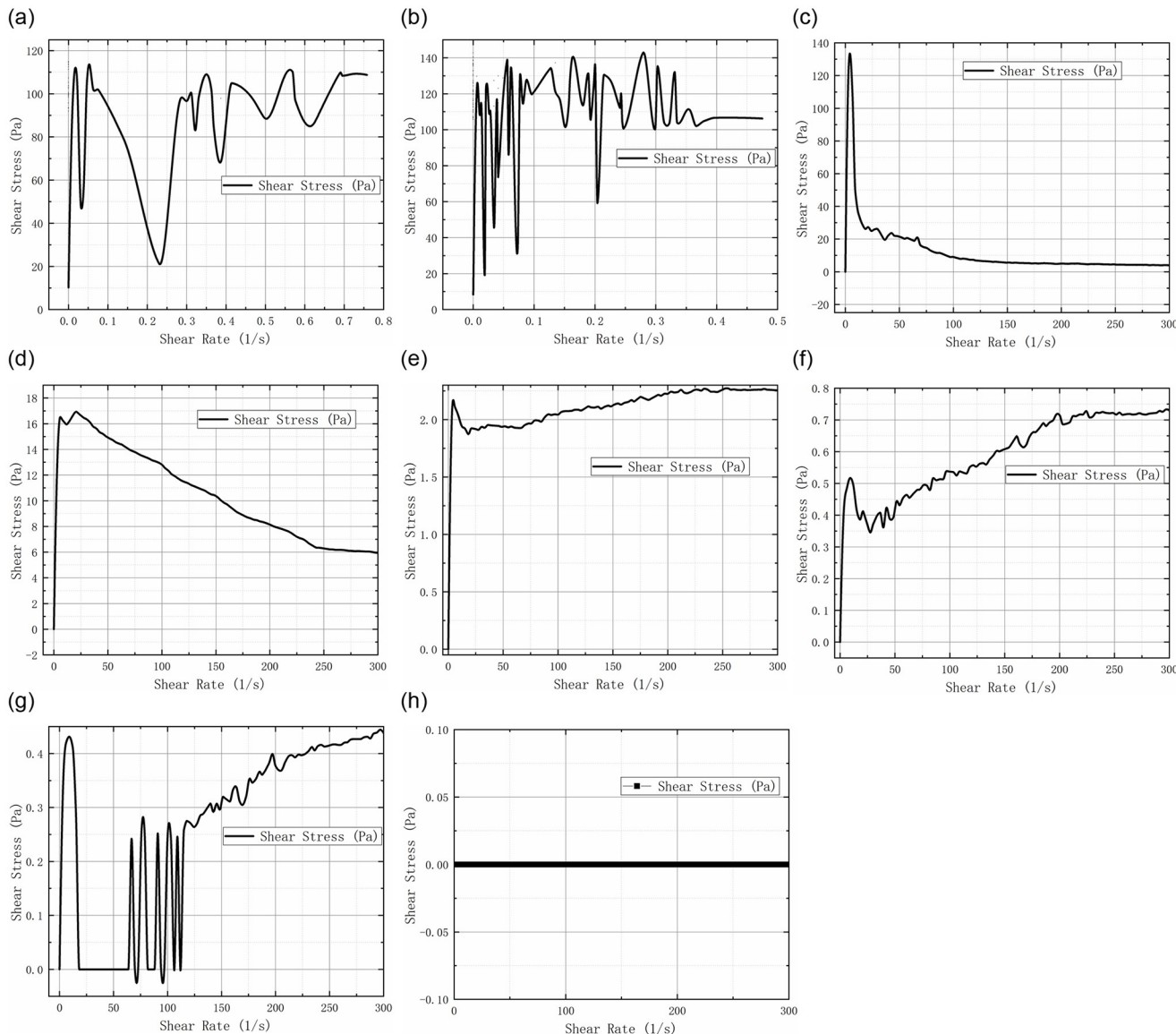

**Fig 4. Rheological test of slurry.** a) First group, b) Second Group, c) Three group, d) Four group, e) Five group, f) Six group, g) Seven group, h) Eight -Ten group.

particles during other seepage processes, it can be found that under different pressure, the permeability of the bentonite slurry was developed, which means high pressure causes more bentonite particles into the sand, which makes the effective permeability decrease.

## 3.5 Microstructure analysis

Microscopic examination of bentonite samples provided visual evidence of structural changes induced by hydration. As water content increased, bentonite particles hydrated and bonded into cohesive clusters, leading to the formation of sponge-like structures. Consolidation tests revealed the vitrification of saturated bentonite, indicating a significant change in its physical properties after hydration.

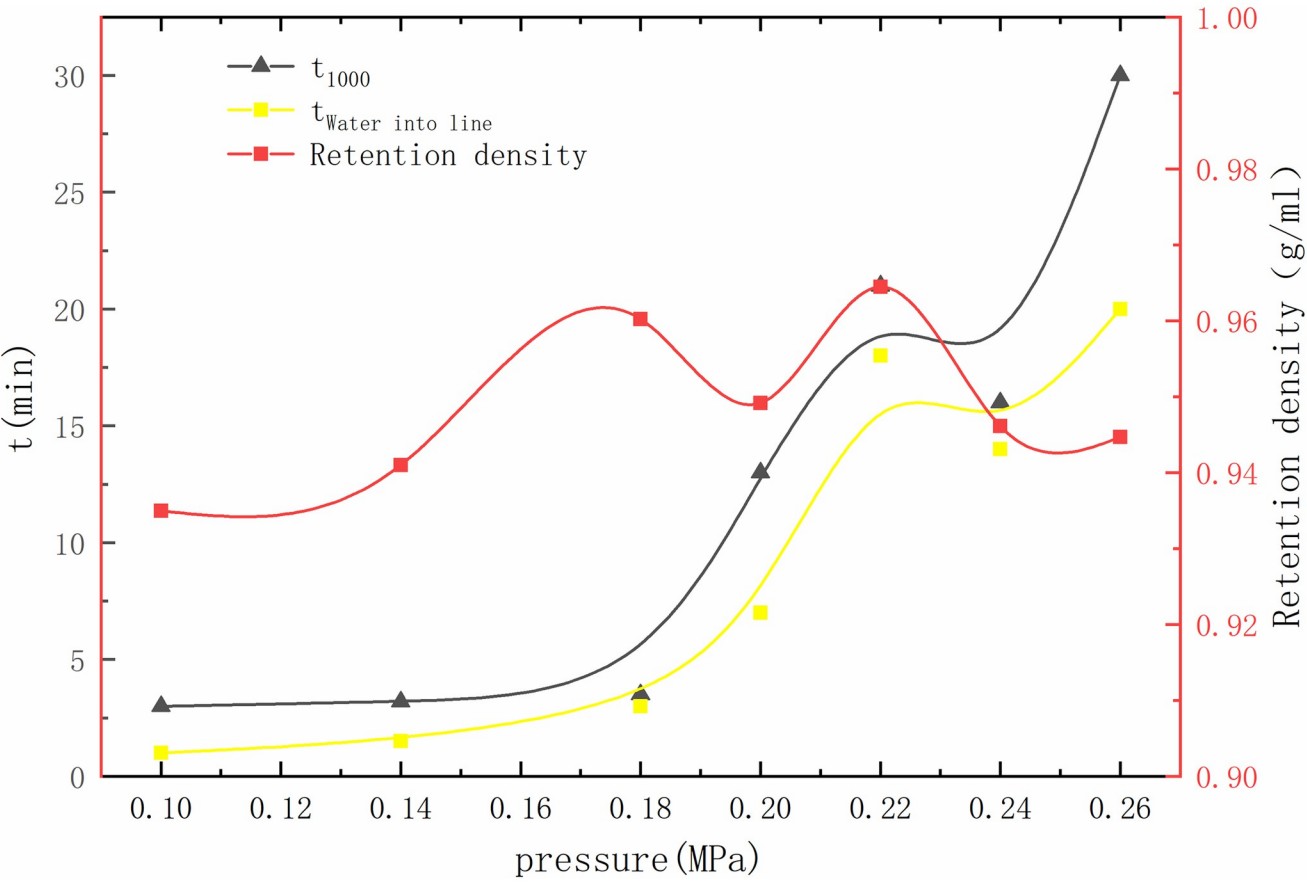

**Fig 5. The permeability test.**

It was found that with the increase of water content of bentonite, the particles gradually hydrated and bonded into clusters, and the bentonite saturated with water content was observed (as shown in Fig 6b), indicating that the particles slowly click together by loose shape (as shown in Fig 6a) and form a sponge-like body. The consolidation test of bentonite with saturated water content under the action of 1kPa for 10 hours is shown in Fig 6(c), and the exact position of the sample is tested and analysed. It can be seen that the bentonite with saturated water content is vitrified, and bentonite is vitrified due to its high density after hydration, film thickness of the combined water and surface vitrification. Fig 6d) shows the drying test of the consolidated soil sample. Comparing Fig 6a), the apparent density of bentonite increases after the wet-dry cycle and the particles are sticky.

## 4. Discussion

The discussion section interprets the experimental findings in light of existing literature and theoretical frameworks. The relationship between water content and the physical properties of bentonite is analyzed, emphasizing the impact of hydration on the mechanical behavior and microstructure of bentonite. The observed changes in rheological properties, permeability, and microstructure are discussed in detail, providing insights into the underlying mechanisms governing bentonite behavior under varying moisture conditions.

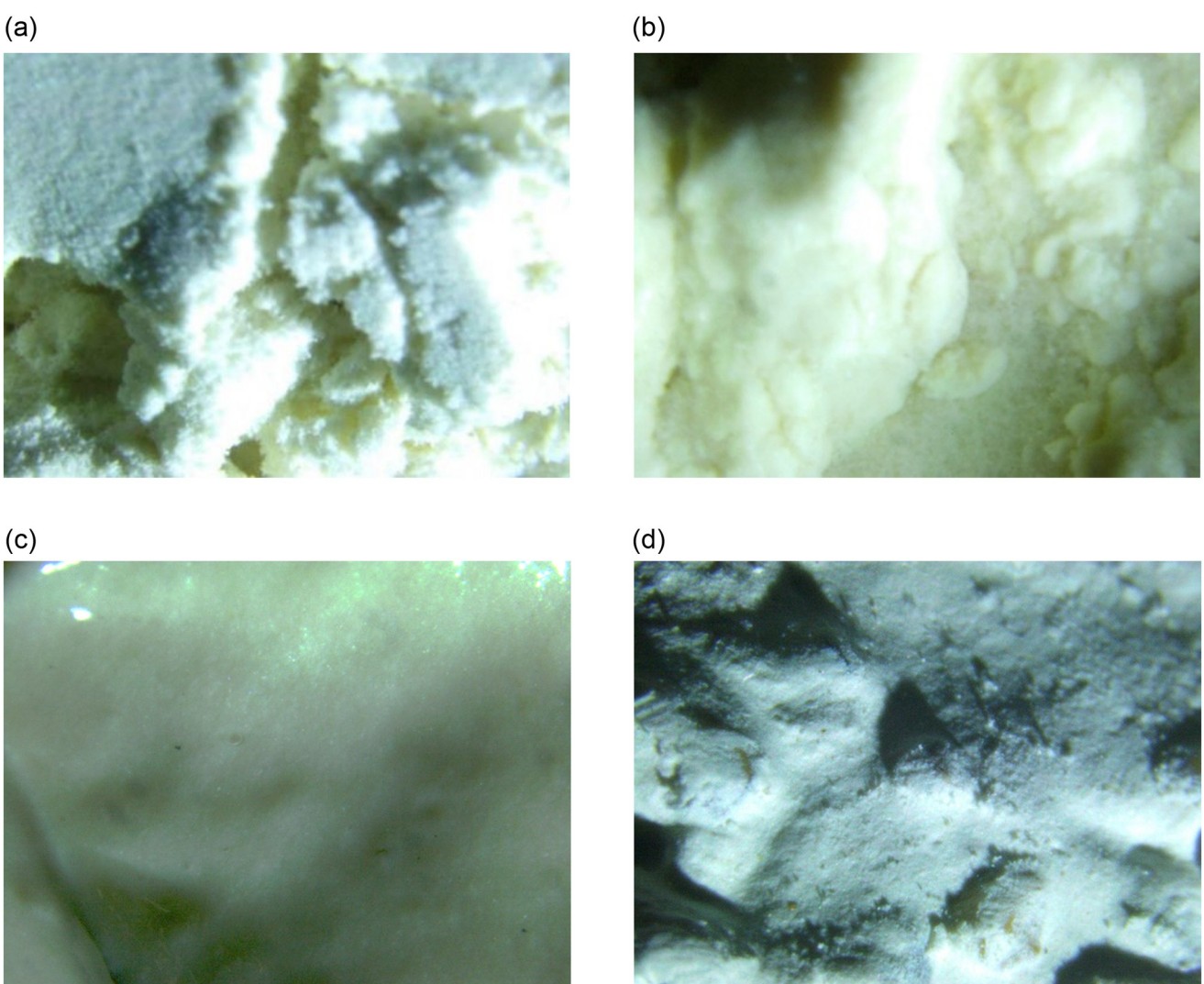

**Fig 6. Results of microstructure analysis.** a) Initial, b) Saturated moisture content state, c)After osmotic consolidation, d) Dry after consolidation.

It is possible to represent the answer (Azouz K B, 2016) of the material to a stress ramp by the following equation(Dupuis et al., 2014)

$$\sigma = \left(1 - \exp\left(-\left(\frac{\gamma}{\gamma_c}\right)^a\right)\right)(\sigma_y + k\dot{\gamma}^n) \tag{1}$$

where $\gamma$, $\gamma$ and $\sigma$ are respectively the shear strain, the shear rate and the shear stress. These three quantities are experimental data. In our experiments, $\sigma$ is imposed, $\gamma$ and $\dot{\gamma}$ are measured on the rheometer. $\gamma$ can also be calculated from the values of $\dot{\gamma}$ and of time t. For large shear strain, Eq (1) reduces to Herschel–Bulkley model [12, 13]

$$\sigma = \sigma_y + k\dot{\gamma}^n \tag{2}$$

where $\sigma_y$ is the yield stress, k is the consistency and n is the flow index. For small shear strains,

Eq (1) reduces to

$$\sigma \approx \frac{\sigma_y}{\gamma_c{}^a}\gamma^a = G_a\gamma^a \tag{3}$$

where $\gamma_c$ is a critical strain corresponding to the transition between a solid-like and a liquid-like behaviour, a is an additional parameter and $G_a = \frac{\sigma_y}{\gamma_c^a}$ a appears as a "pseudo" shear modulus. A value of a close to 1 means that the material behaves as an elastic solid with a stress proportional to the strain.

From Eq (1), it is evident that the rheological properties of bentonite, particularly shear strength, are correlated with shear rate. In this experiment, results obtained at a fixed shear rate indicate that the apparent dynamic viscosity decreases as the strain rate increases, displaying shear-thinning characteristics [14, 15]. This behavior is observed across various pore pressure ratios. However, post-liquefaction, the apparent dynamic viscous strain rate curve steepens with decreasing pore pressure ratio, suggesting a shift in the traditional hydration of bentonite mud, impacting the cohesiveness of the sandy soil mixture.

Combining rheological and permeability tests reveals that with increasing moisture content, the rheological properties of the soil gradually increase. This is attributed to a significant number of bentonite particles entering a suspended state, causing a sharp drop in the cohesion between expansive soil particles and leading to the dispersion of the soil structure. Ultimately, a large number of particles migrate to the bottom sand layer, with the critical pressure between 0.18–0.22 MPa. When the pore pressure reaches around 0.2 MPa, saturated bentonite loses its structure, resulting in a rapid increase in rheological properties.

Microstructure testing of soil at different moisture levels indicates that as moisture content increases, the rheological shear performance of the soil decreases. The primary reason for the increased probability of permeating other materials is the loss of stability in the soil structure and cohesion between particles. Compared to other clay particle structures, where cohesion is lost due to the hydrophobicity of organic matter between particles, the hydrophilic nature of bentonite particles causes rapid expansion, resulting in a sponge-like structure. This active disruption of the soil structure by particle expansion leads to a significant decrease in rheological shear performance. Thus, the relationship between moisture content and soil behavior is such that, as moisture content increases, bentonite particles cause the rupture of the soil structure due to their own expansion, leading to a sharp decline in rheological shear performance. This, in turn, results in a further decrease in cohesion between particles, significantly increasing the ability of dispersed particles to permeate other substances. Ultimately, this leads to the failure of the bentonite protective layer or the collapse of the bentonite cushion layer.

## 5. Conclusion

In this investigation, a comprehensive series of tests was conducted on bentonite samples with varying moisture content to assess their density, free expansion rate, rheological properties, permeability, and microscopic characteristics. The results reveal an initial increase in bentonite density with rising moisture content, followed by a subsequent decline after reaching saturation due to hydration-induced expansion, leading to increased porosity. Furthermore, the expansion rate and shear strength of the particles exhibited an upward trend with increasing water content, albeit with a notable softening tendency emerging at a moisture content threshold of approximately 50%. This suggests a pivotal role of moisture content in modulating the mechanical behavior of bentonite, characterized by a promoting effect prior to saturation, succeeded by a softening trend thereafter. Additionally, our analysis of permeability and microscopic observations elucidates the progressive hydration and structural degradation of

bentonite particles under varying hydrostatic pressures. Of particular interest is the manifestation of expansive swelling in high montmorillonite-content bentonite upon water contact, leading to a disruption of its framework and consequent reduction in rheological shear forces, coupled with increased particle migration—a phenomenon divergent from the behavior observed in conventional clay matrices.

## Supporting information

**S1 File. Original data and plot for Fig 1 in the article.**
(ZIP)

**S2 File. Original data and plot for Fig 2 in the article.**
(PNG)

**S3 File. Original data and plot for Fig 3 in the article.**
(PNG)

**S4 File. Original data and plot for Fig 4 in the article.**
(ZIP)

**S5 File. Original data and plot for Fig 5 in the article.**
(PNG)

**S6 File. Original data and plot for Fig 6 in the article.**
(ZIP)

**S1 Table. Original data and plot for Table 1 in the article.**
(PNG)

**S2 Table. Original data and plot for Table 2 in the article.**
(PNG)

## Acknowledgments

We thank Henan Provincial Engineering Laboratory for Civil Disaster Prevention and Mitigation for providing equipment support for this experiment.

## Author Contributions

**Conceptualization:** Yanzhao Yuan.

**Data curation:** Chao Zheng.

**Funding acquisition:** Chao Zheng.

**Project administration:** Yanzhao Yuan.

**Resources:** Yanzhao Yuan.

**Supervision:** Chao Zheng.

**Writing – original draft:** Yanzhao Yuan.

**Writing – review & editing:** Yanzhao Yuan.

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
