## [Decision Letter · Decision Letter 0]

22 Jan 2024

PONE-D-24-00121Study on the effect of water content on physical properties of bentonitePLOS ONE

Dear Dr.Yan-zhao Yuan,

Thank you for submitting your manuscript to PLOS ONE. After careful consideration, we feel that it has merit but does not fully meet PLOS ONE’s publication criteria as it currently stands. Therefore, we invite you to submit a revised version of the manuscript that addresses the points raised during the review process.

We look forward to receiving your revised manuscript.

Kind regards,

Ajaya Bhattarai

Academic Editor

PLOS ONE

Journal Requirements:

2. In the online submission form, you indicated that your data is available only on request from a third party. Please note that your Data Availability Statement is currently missing [the name of the third party contact or institution / contact details for the third party, such as an email address or a link to where data requests can be made]. Please update your statement with the missing information.

“The authors disclosed receipt of the following financial support for this article's research, authorship, and publication. This work is financially supported by the General project of the Natural Science Foundation of Henan Province（212300410327）”

Reviewers' comments:

Reviewer's Responses to Questions

**Comments to the Author**

1. Is the manuscript technically sound, and do the data support the conclusions?

Reviewer #1: Yes

Reviewer #2: Yes

2. Has the statistical analysis been performed appropriately and rigorously? 

Reviewer #1: Yes

Reviewer #2: Yes

3. Have the authors made all data underlying the findings in their manuscript fully available?

Reviewer #1: Yes

Reviewer #2: No

4. Is the manuscript presented in an intelligible fashion and written in standard English?

Reviewer #1: Yes

Reviewer #2: No

5. Review Comments to the Author

Reviewer #1: Thank you for submitting your research article on the effects of water content on the properties of expansive soils, specifically focusing on bentonite in the context of Longtan Temple area in Chengdu. While the topic is indeed intriguing and potentially valuable to the field of geotechnical engineering, there are several aspects that require clarification and elaboration for the manuscript to meet the standards of a rigorous scientific publication.

1.On line 14, it is mentioned that the study investigates “properties of bentonite.” To enhance clarity and precision, kindly specify which particular physical properties of bentonite are being studied under varying moisture conditions. Examples could include swelling potential, shrinkage, Atterberg limits, or permeability, among others.

Regarding data presentation:

2.On line 21, the term “w saturated” has been used without a clear definition. For readers to fully comprehend the results, please define this term explicitly. It typically refers to the water content at saturation point where all void spaces within the soil particles are filled with water. Please provide a detailed explanation of how this saturation condition was achieved in your experiments and present the corresponding measurements or observations.

In the methodology section:

3.By line 85, the experimental procedures seem incomplete or inadequately described. A more detailed account of the experimental steps would greatly benefit the reproducibility of your study. Please elaborate on the specific methods employed to control and measure the water content, the techniques used for subjecting the bentonite samples to various saturation levels, and the instruments utilized to assess changes in the relevant physical properties.

4.By line 196, the summary does not appear to offer a clear breakdown of the findings related to the “w saturated” condition. In your revised conclusion, please ensure that you draw explicit conclusions about the behavior of bentonite when it reaches its saturated state. This should include a concise summary of how saturation influences the key physical characteristics observed in your experiments, any thresholds identified, and the implications these findings have for mitigating hazards in expansive soil foundations and roadbeds.

I encourage you to address these points thoroughly in your revision, ensuring that each critical aspect of your research is well-defined and supported by the presented data. Your work can significantly contribute to the understanding and management of issues associated with expansive soils once these revisions are made.

Reviewer #2: Based on the research results on the physical properties of bentonite in Nanyang, China, under different water contents, this paper will focus on the fundamental physical properties of bentonite and use a rheometer to measure the fluid properties of bentonite and use a stereomicroscope to comprehensively analyse the influence of water contents on physical properties of bentonite under different water contents. The overall content of the paper is relatively detailed, logical and fluent, and its research object, content and method are reasonable. But there are still the following problems:

1. Research background and literature review: The introduction section of the paper provides an overview of existing research, but lacks in-depth analysis of relevant research and existing literature. Suggest the author to expand the literature review section and discuss in detail the findings of previous studies and how they are relevant to current research.

2. When discussing the effects of different water contents on the physical properties of bentonite, the paper did not fully combine theoretical models or existing theories to explain the observed phenomena. For example, the discussion on how the rheological properties, permeability, and microstructure changes of bentonite are related to its chemical and physical properties at different water contents is not in-depth enough. It is recommended that the author refer more to and cite relevant theories when analyzing the results, in order to provide more in-depth and comprehensive theoretical explanations.

3. When describing the experimental process, the paper lacks detailed information about the experimental conditions and implementation steps, which can affect the repeatability and verifiability of the experimental results. For example, in the free expansion test and rheological property test, the preparation method of the sample, the environmental conditions of the experiment (such as temperature and humidity), and the specific steps of the experimental operation were not detailed. In order to improve the reliability and reproducibility of the research, the author should provide a more detailed description of the experimental methods.

4. The paper lacks sufficient statistical analysis to verify the significance and reliability of the data when presenting experimental results. For example, have repeated experiments been conducted to ensure consistency of results? Have statistical methods been used to analyze differences in data variability and significance? These are key elements to ensure the rigor of scientific research. The author should include relevant statistical analysis, such as mean, standard deviation, confidence interval, etc., when presenting the data to improve the credibility of the study.

5. Although the paper has studied the physical properties of bentonite at different water contents, there is a lack of sufficient comparative analysis with existing research. For example, the findings and conclusions in a paper should be compared with previous research to demonstrate how new research extends or challenges existing knowledge. Suggest the author to add a comparative analysis with existing literature in the discussion section, emphasizing the innovation and scientific contribution of this study。

6. The paper lacks sufficient discussion on the experimental scope and possible limitations. Any scientific research has its limitations, such as sample selection, limitations in experimental conditions, etc., which may affect the universality of results Performance and scope of application. In the paper, it seems that these potential limitations have not been fully identified and discussed. For example, factors such as the source of bentonite samples, particle size distribution, and variability in chemical composition may affect the experimental results. Suggest the author to discuss these limitations in detail in the discussion section and explain their potential impact on the research findings. Meanwhile, it is also possible to propose how to overcome these limitations in future work to enhance the broad applicability and depth of research.

Considering the above problems, it is recommended to the author make a major revision.

6. PLOS authors have the option to publish the peer review history of their article (what does this mean?). If published, this will include your full peer review and any attached files.

Reviewer #1: No

Reviewer #2: No

---

## [Author Response · Author response to Decision Letter 0]

13 Mar 2024

Response to Reviewers:

We would like to express our gratitude to the reviewers for their valuable feedback and constructive comments on our manuscript titled "Effects of Water Content on the Properties of Bentonite in Expansive Soils"

To address the concerns raised by Reviewer #1:

1. Clarification of Investigated Properties: We appreciate the suggestion to specify the physical properties of bentonite studied under varying moisture conditions. In response, we have provided a detailed explanation of the properties investigated, including swelling potential, shrinkage, Atterberg limits, and permeability, in the revised manuscript.

2.Definition of "w saturated": Thank you for highlighting the need for a clear definition of the term "w saturated." We have added an explicit definition in the manuscript and explained how the saturation condition was achieved in our experiments.

3.Elaboration of Experimental Procedures: We acknowledge the need for a more detailed description of the experimental procedures. In the revised manuscript, we have provided a comprehensive account of the methods employed to control and measure water content, subject bentonite samples to various saturation levels, and assess changes in relevant physical properties.

4.Improved Summary of Findings: We have revised the conclusion to offer a clearer breakdown of the findings related to the saturated condition of bentonite. This includes explicit conclusions about the behavior of bentonite when saturated, thresholds identified, and the implications for mitigating hazards in expansive soil foundations and roadbeds.

To address the concerns raised by Reviewer #2:

1.Expanded Literature Review: We have expanded the literature review section to provide a more in-depth analysis of relevant research and discuss how previous studies are relevant to our research.

2.In-depth Theoretical Analysis: We have enhanced the discussion on the effects of different water contents on the physical properties of bentonite by incorporating relevant theoretical models and existing theories to explain observed phenomena more comprehensively.

Detailed Experimental Description: We have provided more detailed information about the experimental conditions and implementation steps to improve the repeatability and verifiability of our experimental results.

Statistical Analysis: We have included relevant statistical analysis, such as mean, standard deviation, and confidence interval, when presenting the data to improve the credibility of our study.

3.Comparative Analysis with Existing Literature: In the discussion section, we have added a comparative analysis with existing literature to highlight the innovation and scientific contribution of our study.

Discussion of Experimental Scope and Limitations: We have discussed the potential limitations of our study, including sample selection and experimental conditions, and proposed ways to overcome these limitations in future work to enhance the broad applicability and depth of our research.

We believe that these revisions address the concerns raised by the reviewers and significantly improve the quality and rigor of our manuscript. Thank you for the opportunity to revise and resubmit our work. We look forward to your feedback on the revised manuscript.

Sincerely, 

Yuan Yanzhao

---

## [Decision Letter · Decision Letter 1]

11 Apr 2024

PONE-D-24-00121R1Study on the effect of water content on physical properties of bentonitePLOS ONE

Dear Dr. Yuan,

Thank you for submitting your manuscript to PLOS ONE. After careful consideration, we feel that it has merit but does not fully meet PLOS ONE’s publication criteria as it currently stands. Therefore, we invite you to submit a revised version of the manuscript that addresses the points raised during the review process.

We look forward to receiving your revised manuscript.

Kind regards,

Ajaya Bhattarai

Academic Editor

PLOS ONE

Journal Requirements:

Additional Editor Comments:

Please revise the manuscript according to the comments of reviewer 3.

Reviewers' comments:

Reviewer's Responses to Questions

**Comments to the Author**

1. If the authors have adequately addressed your comments raised in a previous round of review and you feel that this manuscript is now acceptable for publication, you may indicate that here to bypass the “Comments to the Author” section, enter your conflict of interest statement in the “Confidential to Editor” section, and submit your "Accept" recommendation.

Reviewer #1: All comments have been addressed

Reviewer #3: (No Response)

2. Is the manuscript technically sound, and do the data support the conclusions?

Reviewer #1: Yes

Reviewer #3: (No Response)

3. Has the statistical analysis been performed appropriately and rigorously? 

Reviewer #1: (No Response)

Reviewer #3: (No Response)

4. Have the authors made all data underlying the findings in their manuscript fully available?

Reviewer #1: Yes

Reviewer #3: (No Response)

5. Is the manuscript presented in an intelligible fashion and written in standard English?

Reviewer #1: Yes

Reviewer #3: No

6. Review Comments to the Author

Reviewer #1: authors have adequately addressed my comments raised in a previous round of review and I feel that this manuscript is now acceptable for publication.

Reviewer #3: 1. The Abstract should be re-written.

2. Please ask an English native speaker to review your paper as the English of the current version is poor.

3. Please add the number of all equations in the manuscript.

4. The figures are not clear, please adjust them through the manuscript.

5. The Conclusion is too long, which only needs to list the important findings in the manuscript.

7. PLOS authors have the option to publish the peer review history of their article (what does this mean?). If published, this will include your full peer review and any attached files.

Reviewer #1: No

Reviewer #3: No

---

## [Author Response · Author response to Decision Letter 1]

24 Apr 2024

Dear editor

Thank you for taking the time to review our manuscript. We appreciate your valuable feedback and have made the necessary revisions to address the issues raised. Below are the responses to each of your points:

Abstract Revision:

We have revised the abstract to enhance clarity and coherence. The revised abstract now provides a concise summary of the study's objectives, methods, key findings, and implications. We believe that these changes will improve the overall readability and effectiveness of the abstract.

English Language Review:

We acknowledge the importance of ensuring the clarity and accuracy of language in the manuscript. To address this concern, we have consulted with a native English speaker to review the paper and make necessary corrections to improve language quality. We are confident that these revisions have enhanced the readability and professionalism of the manuscript.

Equation Numbering:

We have added numbering to all equations in the manuscript for ease of reference. This enhancement will facilitate readers in identifying and referring to specific equations throughout the document.

Figure Clarity Adjustment:

We have carefully reviewed all figures in the manuscript and made necessary adjustments to improve clarity. These adjustments include enhancing resolution, adjusting size, and ensuring consistency in formatting across all figures. We believe that these modifications will enhance the visual presentation of our findings.

Conclusion Length Reduction:

We have shortened the conclusion section to focus solely on summarizing the important findings of the study. By eliminating unnecessary details, the revised conclusion now provides a concise overview of the key insights derived from our research.

Overall, we believe that these revisions have significantly improved the quality and effectiveness of our manuscript. We sincerely appreciate your constructive feedback, which has helped us enhance the clarity, coherence, and professionalism of our work. Please feel free to reach out if you have any further questions or require additional information.

Thank you once again for your time and valuable input.

Best regards,

Yuan Yanzhao

Henan university of urban contrarian

---

## [Editor Report · Decision Letter 2]

26 Apr 2024

Study on the effect of water content on physical properties of bentonite

PONE-D-24-00121R2

Dear Dr. Yuan,

We’re pleased to inform you that your manuscript has been judged scientifically suitable for publication and will be formally accepted for publication once it meets all outstanding technical requirements.

Kind regards,

Ajaya Bhattarai

Academic Editor

PLOS ONE

Additional Editor Comments (optional):

The revised manuscript looks good.
---

## [Editor Report · Acceptance letter]

14 Jun 2024

PONE-D-24-00121R2 

PLOS ONE

Dear Dr. Yuan, 

I'm pleased to inform you that your manuscript has been deemed suitable for publication in PLOS ONE. Congratulations! Your manuscript is now being handed over to our production team.

Kind regards, 

on behalf of

Dr. Ajaya Bhattarai 

Academic Editor

PLOS ONE